# Development of a Novel Anti-CD44 Variant 4 Monoclonal Antibody C$_{44}$Mab-108 for Immunohistochemistry

**Hiroyuki Suzuki** [1,2,*] **, Tomohiro Tanaka** [1] **, Nohara Goto** [1] **, Mika K. Kaneko** [1,2] **and Yukinari Kato** [1,2,*]

[1]  Department of Molecular Pharmacology, Tohoku University Graduate School of Medicine, 2-1 Seiryo-Machi, Aoba-Ku, Sendai 980-8575, Japan

[2]  Department of Antibody Drug Development, Tohoku University Graduate School of Medicine, 2-1 Seiryo-Machi, Aoba-Ku, Sendai 980-8575, Japan

*  Correspondence: hiroyuki.suzuki.b4@tohoku.ac.jp (H.S.); yukinari.kato.e6@tohoku.ac.jp (Y.K.); Tel.: +81-29-853-3944 (H.S. & Y.K.)

**Abstract:** CD44 has been known as a marker of tumor-initiating cells, and plays pro-tumorigenic functions in many cancers. The splicing variants play critical roles in the malignant progression of cancers by promoting stemness, cancer cell invasion or metastasis, and resistance to chemo- and radiotherapy. To understand each CD44 variant (CD44v) function is essential to know the property of cancers and the establishment of the therapy. However, the function of the variant 4-encoded region has not been elucidated. Therefore, specific monoclonal antibodies (mAbs) against variant 4 are indispensable for basic research, tumor diagnosis, and therapy. In this study, we established anti-CD44 variant 4 (CD44v4) mAbs by immunizing mice with a peptide containing the variant 4-encoded region. We next performed flow cytometry, western blotting, and immunohistochemistry to characterize them. One of the established clones (C$_{44}$Mab-108; IgG$_1$, kappa) reacted with CD44v3-10-overexpressed Chinese hamster ovary-K1 cells (CHO/CD44v3-10). The $K_D$ of C$_{44}$Mab-108 for CHO/CD44 v3-10 was $3.4 \times 10^{-7}$ M. In western blot analysis, C$_{44}$Mab-108 detected CD44v3-10 in the lysate of CHO/CD44v3-10 cells. Furthermore, C$_{44}$Mab-108 stained formalin-fixed paraffin-embedded (FFPE) oral squamous carcinoma tissues in immunohistochemistry. These results indicated that C$_{44}$Mab-108 is useful to detect CD44v4 in immunohistochemistry using FFPE tissues.

**Keywords:** CD44; CD44 variant 4; monoclonal antibody; flow cytometry; immunohistochemistry

## 1. Introduction

CD44 is a type I transmembrane glycoprotein, which is widely distributed in normal tissues [1]. CD44 consists of several exons. CD44 standard isoform (CD44s) is the shortest isoform of CD44 (85–95 kDa) which is produced by assembling the first five and the last five constant region exons [2]. The middle variant exons (v1–v10) can be alternatively spliced and assembled with the first five and the last five exons of CD44s. They are defined as CD44 variant isoforms (CD44v) [3]. It attracted considerable interest when it was discovered that CD44v induced metastatic properties in tumor cells [4]. A growing body of evidence suggests the importance of CD44v in the malignant progression of tumors [5], as well as cancer-initiating properties [6].

Both CD44s and CD44v can bind to hyaluronic acid (HA) and are involved in cell adhesion, proliferation, and migration [7,8]. CD44s is widely expressed in normal tissues and plays important roles in hematopoiesis, the immune system, and organogenesis [9]. CD44v plays critical roles in the promotion of tumor invasion, metastasis, stemness, and resistance to chemo- and radiotherapy [10,11]. CD44v3-10 can bind to heparin-binding epidermal growth factor-like growth factor (HB-EGF) and fibroblast growth factors (FGFs) via v3-encoded region, and functions as a co-receptor of receptor tyrosine kinases [12]. Moreover, the v6-encoded region is essential for the recruitment of hepatocyte growth factor (HGF) to its receptor, c-MET [13]. In addition, the v8-10-encoded region confers

oxidative stress resistance [14]. Therefore, understanding the function of each variant is essential to identify the properties of carcinomas. However, the function of the variant 4-encoded region has not been elucidated. Therefore, specific antibodies against CD44v4 are indispensable for basic research, tumor diagnosis, and therapy.

We previously established the novel anti-CD44 mAbs, $C_{44}$Mab-5 (IgG$_1$, kappa) [15], and $C_{44}$Mab-46 (IgG$_1$, kappa) [16] using Cell-Based Immunization and Screening (CBIS) method and immunization of CD44v3-10 ectodomain, respectively. Both $C_{44}$Mab-5 and $C_{44}$Mab-46 have epitopes in the first five exons-encoding sequences [17–19]. Therefore, they can recognize both CD44s and CD44v (pan-CD44). Furthermore, they showed high sensitivity for flow cytometry and immunohistochemical analysis in oral [15] and esophageal tumors [16]. We have also investigated the antitumor effects in mouse xenograft models of oral squamous cell carcinomas (OSCC) [20].

In this study, we developed a novel anti-CD44v4 mAb, $C_{44}$Mab-108 (IgG$_1$, kappa) by peptide immunization of the v4-encoded region, and evaluated its applications, including flow cytometry, western blotting, and immunohistochemical analyses.

## 2. Materials and Methods

### 2.1. Cell Lines

Chinese hamster ovary (CHO)-K1 and P3X63Ag8U.1 (P3U1, mouse multiple myeloma) cell lines were obtained from the American Type Culture Collection (ATCC, Manassas, VA, USA). Esophageal squamous cell carcinoma cell lines, KYSE70 and KYSE770 were obtained from the Japanese Collection of Research Bioresources (Osaka, Japan). CD44s open reading frame (ORF) was amplified from LN229 cDNA using HotStar HiFidelity Polymerase Kit (Qiagen Inc., Hilden, Germany). CD44v3-10 ORF was provided by the RIKEN BRC through the National Bio-Resource Project of the MEXT, Japan. CD44s and CD44v3-10 ORFs were subcloned into pCAG-Ble-ssPA16 vector possessing signal sequence and the PA16 tag (GLEGGVAMPGAEDDVV) [15,21–24], which is detected by NZ-1 [25–35]. CHO/CD44s and CHO/CD44v3-10 were established by transfecting pCAG-Ble/PA16-CD44s and pCAG-Ble/PA16-CD44v3-10 into CHO-K1 cells using a Neon transfection system (Thermo Fisher Scientific, Inc., Waltham, MA, USA).

CHO-K1 and P3U1 cells were cultured in Roswell Park Memorial Institute (RPMI)-1640 medium (Nacalai Tesque, Inc., Kyoto, Japan), supplemented with 10% heat-inactivated fetal bovine serum (FBS; Thermo Fisher Scientific, Inc.), 100 µg/mL streptomycin, 100 U/mL penicillin, 0.25 µg/mL amphotericin B (Nacalai Tesque, Inc.), and 50 µg/mL plasmocin prophylactic (InvivoGen, San Diego, CA, USA). KYSE70 and KYSE770 were cultured in Dulbecco's Modified Eagle Medium (DMEM; 4.5 g/L glucose), containing L-Gln and without sodium pyruvate (Nacalai Tesque, Inc.), 10% FBS, 100 U/mL of penicillin, 100 µg/mL streptomycin, 0.25 µg/mL amphotericin B, and 50 µg/mL plasmocin prophylactic. All the cells were cultured in a humidified incubator at 37 °C with 5% $CO_2$.

### 2.2. Peptides

Sigma-Aldrich (St. Louis, MO, USA) synthesized a partial sequence of the human CD44 variant 4 ($_{273}$DHTKQNQDWTQWNPSHSNP$_{291}$) plus C-terminal cysteine. Subsequently, the keyhole limpet hemocyanin (KLH) was conjugated at the C-terminus of the peptide. CD44 peptides, which cover the extracellular domain of CD44v3-10, were described previously [17].

### 2.3. Hybridoma Production

The female BALB/c mice were purchased from CLEA Japan (Tokyo, Japan). All animal experiments were conducted according to the guidelines and regulations to minimize animal suffering and distress in the laboratory. The Animal Care and Use Committee of Tohoku University (Permit number: 2019NiA-001) approved animal experiments. The mice were housed under specific pathogen-free conditions and monitored daily for health during the duration of the experiment. To develop mAbs against CD44v4, we intraperitoneally

immunized mice with the KLH-conjugated CD44v4 peptide (100 µg) plus an adjuvant (Imject Alum; Thermo Fisher Scientific Inc.). The procedure included three additional weekly immunization (100 µg/mouse). A final booster injection (100 µg/mouse) was performed two days before harvesting spleen cells. The harvested spleen cells were subsequently fused with P3U1 cells, using PEG1500 (Roche Diagnostics, Indianapolis, IN, USA). The hybridomas were selected in the presence of hypoxanthine, aminopterin, and thymidine (HAT; Thermo Fisher Scientific Inc.). The supernatants were screened using enzyme-linked immunosorbent assay (ELISA) with the CD44v4 peptide [1 µg/mL in phosphate-buffered saline (PBS)], followed by flow cytometry using CHO/CD44v3-10 and CHO-K1 cells. CHO/CD44v3-10-reactive and parental CHO-K1-non-reactive supernatants were determined to be positive for CD44v3-10. To establish single clones, limiting dilution was performed.

## 2.4. ELISA

The synthesized peptide (DHTKQNQDWTQWNPSHSNP), which is included in CD44v4 plus C-terminal cysteine, was immobilized at a concentration of 1 µg/mL in PBS on Nunc Maxisorp 96 well immunoplates (Thermo Fisher Scientific Inc.) for 30 min at 37 °C. This peptide might be dimerized using the disulfide bond. The plates were washed with 0.05% Tween 20 in PBS (PBST; Nacalai Tesque, Inc.), and the wells were blocked with 1% bovine serum albumin (BSA) in PBST for 30 min at 37 °C. Then, the plates were incubated with the supernatants of hybridomas, followed by peroxidase-conjugated anti-mouse immunoglobulins (1:2000 diluted; Agilent Technologies Inc., Santa Clara, CA, USA). ELISA POD Substrate TMB Kit (Nacalai Tesque, Inc.) was used for enzymatic reactions. The optical density at 655 nm was measured using an iMark microplate reader (Bio-Rad Laboratories, Inc., Berkeley, CA, USA).

In Supplementary Table S1, CD44v3−10 ectodomain [16] was immobilized at a concentration of 1 µg/mL for 30 min at 37 °C. After blocking with 1% BSA in PBST for 30 min at 37 °C, the plates were incubated with the supernatant of $C_{44}$Mab-108, followed by incubation with peroxidase-conjugated anti-mouse immunoglobulins, anti-mouse heavy chains ($IgG_1$, $IgG_{2a}$, $IgG_{2b}$, $IgG_3$, and IgM; SouthernBiotech, Birmingham, AL, USA), or anti-mouse light chains (kappa and lambda; SouthernBiotech). The enzymatic reactions were conducted and the optical density was measured. $C_{44}$Mab-108 isotype was determined by the reactivity of secondary antibodies.

In Supplementary Figure S2, fifty-eight synthesized peptides [17] were immobilized at a concentration of 10 µg/mL for 30 min at 37 °C. After blocking with 1% BSA in PBST for 30 min at 37 °C, the plates were incubated with $C_{44}$Mab-108 (1 µg/mL), followed by incubation with peroxidase-conjugated anti-mouse immunoglobulins. The enzymatic reactions were conducted and the optical density was measured.

## 2.5. Flow Cytometry

The cells were isolated using 0.25% trypsin and 1 mM ethylenediaminetetraacetic acid (Nacalai Tesque, Inc.). The cells were washed with blocking buffer (0.1% BSA in PBS) and treated with $C_{44}$Mab-108 or $C_{44}$Mab-46 for 30 min at 4 °C. Subsequently, the cells were treated with Alexa Fluor 488-conjugated anti-mouse IgG (1:2000; Cell Signaling Technology, Inc., Danvers, MA, USA) for 30 min at 4 °C. Fluorescence data were collected using the SA3800 Cell Analyzer and analyzed using SA3800 software ver. 2.05 (Sony Corporation, Tokyo, Japan) and FlowJo (BD Biosciences, Franklin Lakes, NJ, USA).

## 2.6. Determination of Dissociation Constant ($K_D$) via Flow Cytometry

Serially diluted $C_{44}$Mab-108 was suspended with CHO/CD44v3-10 cells. The cells were further treated with Alexa Fluor 488-conjugated anti-mouse IgG (1:200; Cell Signaling Technology, Inc.). Fluorescence data were collected using BD FACSLyric and analyzed using BD FACSuite software version 1.3 (BD Biosciences). To determine the dissociation

constant ($K_D$), GraphPad Prism 8 (the fitting binding isotherms to built-in one-site binding models; GraphPad Software, Inc., San Diego, CA, USA) was used.

### 2.7. Western Blot Analysis

The cell lysates (10 μg of protein) were prepared using sodium dodecyl sulfate (SDS) sample buffer (Nacalai Tesque, Inc.), and separated on 5–20% polyacrylamide gels (FUJI-FILM Wako Pure Chemical Corporation, Osaka, Japan). The proteins were transferred onto polyvinylidene difluoride (PVDF) membranes (Merck KGaA, Darmstadt, Germany). The membranes were incubated with 10 μg/mL of $C_{44}$Mab-108, 1 μg/mL of an anti-PA16 tag mAb (NZ-1), or 1 μg/mL of anti-β-actin mAb (clone AC-15; Sigma-Aldrich Corp.) in 4% skim milk (Nacalai Tesque, Inc.) in PBS with 0.05% Tween 20. Then, the membranes were incubated with anti-mouse immunoglobulins conjugated with peroxidase (diluted 1:1000; Agilent Technologies, Inc.) for $C_{44}$Mab-108 and anti-β-actin. The anti-rat immunoglobulins (diluted 1:10,000; Sigma-Aldrich Corp.) conjugated with peroxidase were used for NZ-1. Finally, the signals were detected with a chemiluminescence reagent, ImmunoStar LD (FUJIFILM Wako Pure Chemical Corporation) using a Sayaca-Imager (DRC Co. Ltd., Tokyo, Japan).

### 2.8. Immunohistochemical Analysis

The formalin-fixed paraffin-embedded (FFPE) OSCC tissue microarray (Product Code: OR601c, US Biomax Inc., Rockville, MD, USA) and the esophageal tissue microarray (Product Code: BC02011, US Biomax Inc.) were deparaffinized in xylene (Sigma-Aldrich Corp.) and rehydrated. The FFPE OSCC tissue for peptide blocking assay was obtained from Tokyo Medical and Dental University [36]. The tissues were autoclaved in citrate buffer (pH 6.0; Nichirei Biosciences, Inc., Tokyo, Japan) for 20 min for antigen retrieval. After blocking with SuperBlock T20 (Thermo Fisher Scientific, Inc.), the sections were incubated with $C_{44}$Mab-108 (10 μg/mL) and $C_{44}$Mab-46 (1 μg/mL), or without the primary antibody (control) for 1 h at room temperature and then treated with the EnVision+ Kit for mouse (Agilent Technologies Inc.) for 30 min. The color was developed using 3,3′-diaminobenzidine tetrahydrochloride (DAB; Agilent Technologies Inc.) for 2 min. Counterstaining was performed with hematoxylin (FUJIFILM Wako Pure Chemical Corporation). Hematoxylin and eosin (HE) staining (FUJIFILM Wako Pure Chemical Corporation) was performed using consecutive tissue sections. Leica DMD108 (Leica Microsystems GmbH, Wetzlar, Germany) was used to examine the sections and obtain images.

## 3. Results

### 3.1. Development of Anti-CD44v4 mAbs by Peptide Immunization

To develop anti-CD44v4 mAbs, mice were immunized with the KLH-conjugated CD44v4 peptide (Figure 1A). The splenocytes were fused with myeloma P3U1 cells (Figure 1B). The developed hybridomas were subsequently seeded into 96-well plates and cultured for six days. The positive wells for the naked CD44v4 peptide were selected using ELISA, followed by the selection of CHO/CD44v3-10-reactive and parental CHO-K1-non-reactive supernatants using flow cytometry (Figure 1C). After the limiting dilution, 12 clones were established. After several additional screenings, including flow cytometry and immunohistochemistry, an anti-CD44v4 mAb (clone $C_{44}$Mab-108; mouse IgG$_1$, kappa) was finally selected and investigated in this study (Figure 1D and Supplementary Table S1). We confirmed that $C_{44}$Mab-108 recognized only a CD44v4-containing peptide (aa 271–290) among peptides, which cover the extracellular domain of CD44v3-10 [17] (Supplementary Figure S1).

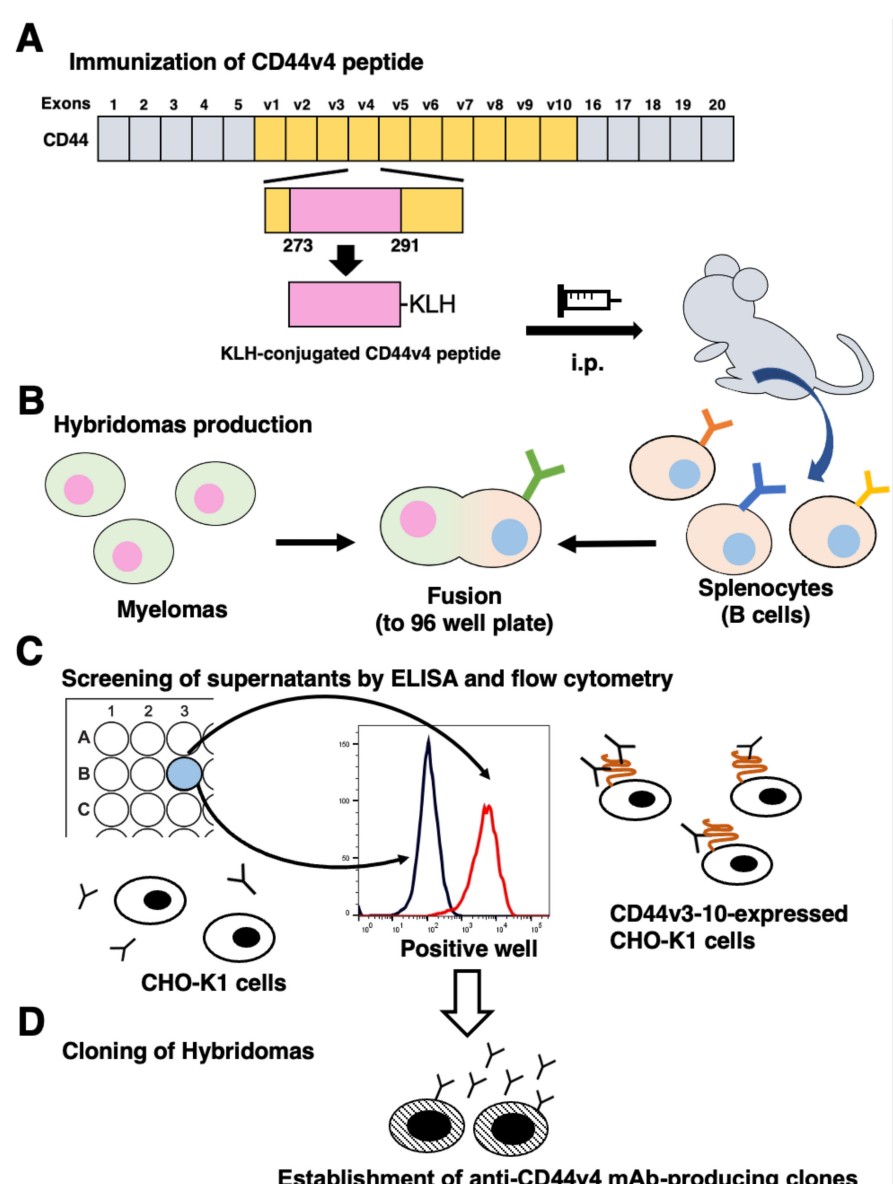

**Figure 1.** A schematic procedure of anti-CD44v4 mAbs production. (**A**) The mice were intraperitoneally immunized with the CD44v4 peptide conjugated with keyhole limpet hemocyanin (KLH). (**B**) The spleen cells were fused with P3U1 cells, and hybridomas were produced. (**C**) The supernatants were subsequently screened using enzyme-linked immunosorbent assay (ELISA) with the CD44v4 peptide, followed by flow cytometry using CHO/CD44v3-10 and CHO-K1 cells. CHO/CD44v3-10-reactive and parental CHO-K1-non-reactive supernatants were determined to be positive for CD44v3-10. (**D**) To establish single clones, limiting dilution was performed. A clone $C_{44}$Mab-108 (IgG$_1$, kappa) was finally established.

### 3.2. Flow Cytometric Analysis

We conducted flow cytometry using $C_{44}$Mab-108 against CHO/CD44s, CHO/CD44v3-10, and CHO-K1 cells. $C_{44}$Mab-108 recognized CHO/CD44v3-10 cells dose-dependently at 10, 1, and 0.1 µg/mL. $C_{44}$Mab-108 did not recognize parental CHO-K1 cells even at 10 µg/mL (Figure 2A). Furthermore, $C_{44}$Mab-108 never recognized CHO/CD44s cells, which were recognized by a previously established anti-pan-CD44 mAb, $C_{44}$Mab-46 [16] at 10 µg/mL (Figure 2B). These results indicated that $C_{44}$Mab-108 specifically recognizes CD44v3-10, but not CD44s by flow cytometry.

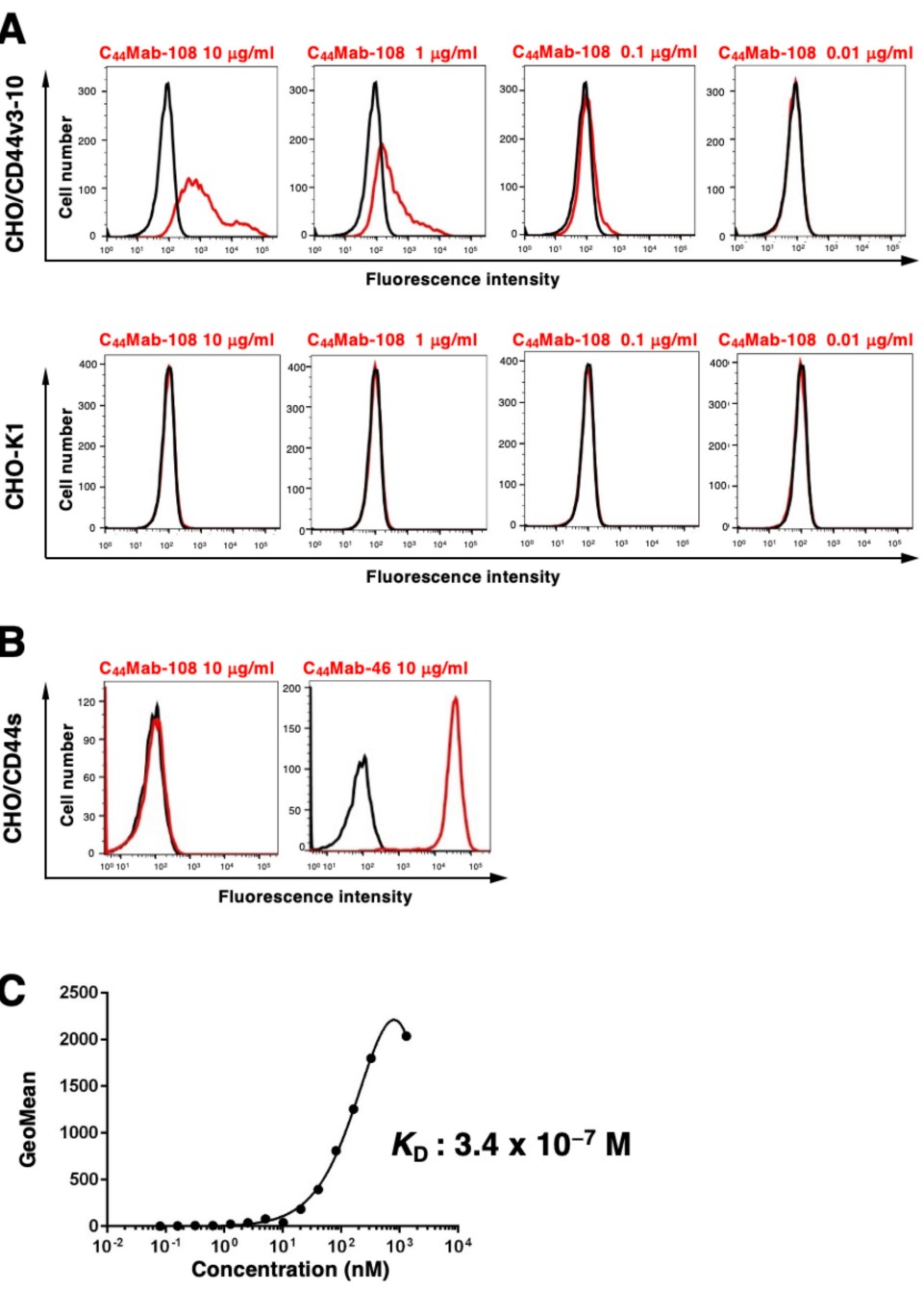

**Figure 2.** Flow cytometry to CD44 expressing cells using $C_{44}$Mab-108. (**A**) The CHO/CD44v3-10 and CHO-K1 cells were treated with 0.01–10 μg/mL of $C_{44}$Mab-108, followed by treatment with Alexa Fluor 488-conjugated anti-mouse IgG. The red lines show the cells with the $C_{44}$Mab-108 treatment. The black lines represent the negative control. (**B**) The CHO/CD44s cells were treated with 10 μg/mL of $C_{44}$Mab-108 and $C_{44}$Mab-46, followed by treatment with Alexa Fluor 488-conjugated anti-mouse IgG. The red lines show the cells with the mAbs treatment. The black lines represent the negative control. (**C**) The CHO/CD44v3-10 were suspended in 100 μL of serially diluted $C_{44}$Mab-108 (0.08–1303 nM). The cells were treated with Alexa Fluor 488-conjugated anti-mouse IgG. The fluorescence data were collected using a BD FACSLyric, following the calculation of the dissociation constant ($K_D$) by GraphPad PRISM 8.

We next determined the $K_D$ of $C_{44}$Mab-108 with CHO/CD44v3-10 cells using flow cytometry. As shown in Figure 2C, the $K_D$ of $C_{44}$Mab-108 for CHO/CD44v3-10 was determined as $3.4 \times 10^{-7}$ M.

### 3.3. Western Blot Analysis

Western blotting was performed to further assess the sensitivity of $C_{44}$Mab-108. The lysates of CHO-K1, CHO/CD44s, and CHO/CD44v3-10 cells were probed. As shown in Figure 3A, $C_{44}$Mab-108 detected CD44v3-10 as a ~120-kDa band. However, $C_{44}$Mab-108 did not detect any bands from the lysates of CHO-K1 and CHO/CD44s cells. An anti-PA16 tag mAb, NZ-1, recognized the lysates from CHO/CD44s (~75 kDa) and CHO/CD44v3-10 (~120 kDa), both of which possess the PA16 tag at their N-terminus. Next, we examined the detection of endogenous CD44v4 using lysates from KYSE70 and KYSE770 cells. As shown in Figure 3B, $C_{44}$Mab-108 could detect CD44v4 as 120 kDa-band from lysates of KYSE770 cells. These results indicated that $C_{44}$Mab-108 can detect exogenous CD44v3-10 and endogenous CD44v4.

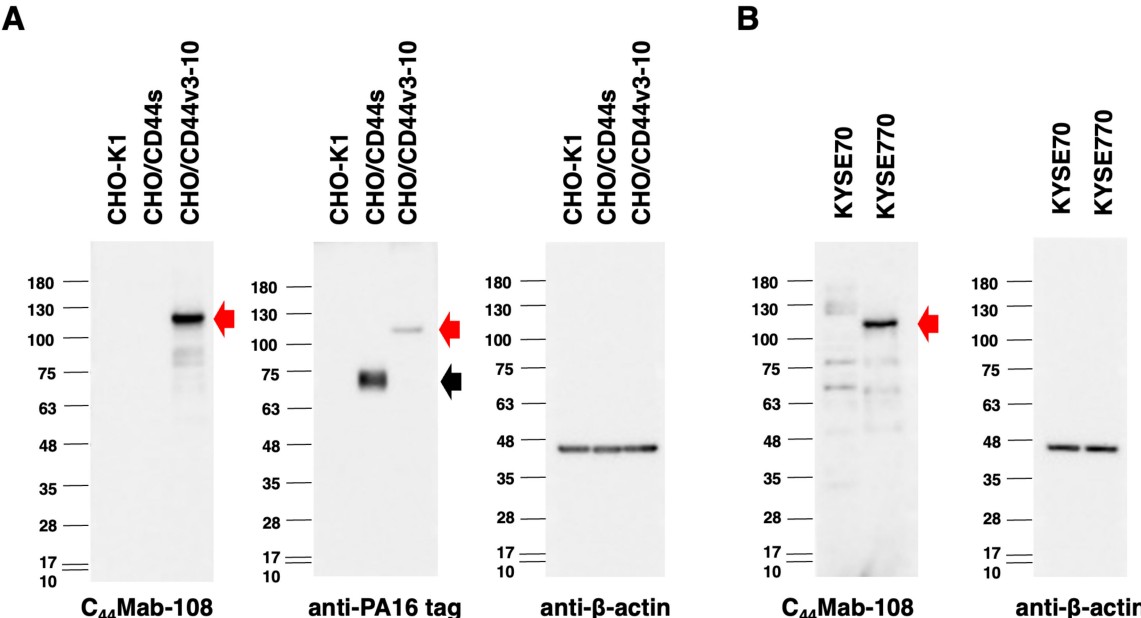

**Figure 3.** Western blotting by $C_{44}$Mab-108. (**A**) The cell lysates of CHO-K1, CHO/CD44s, and CHO/CD44v3-10 (10 µg) were electrophoresed and transferred onto polyvinylidene fluoride (PVDF) membranes. The membranes were incubated with 10 µg/mL of $C_{44}$Mab-108, 1 µg/mL of anti-PA16 tag mAb (NZ-1), and 1 µg/mL of anti-β-actin mAb. Then, the membranes were incubated with anti-mouse immunoglobulins conjugated with peroxidase for $C_{44}$Mab-108 and anti-β-actin. The anti-rat immunoglobulins conjugated with peroxidase was used for NZ-1. The black arrow indicates the CD44s (~75 kDa). The red arrows indicate the CD44v3-10 (~120 kDa). (**B**) The cell lysates of KYSE70 and KYSE770 (10 µg) were electrophoresed and transferred onto PVDF membranes. The membranes were incubated with 10 µg/mL of $C_{44}$Mab-108 and 1 µg/mL of anti-β-actin and subsequently with peroxidase-conjugated anti-mouse immunoglobulins. The red arrow indicates the CD44v4 (~120 kDa).

### 3.4. Immunohistochemical Analysis Using $C_{44}$Mab-108 against OSCC Tissues

To investigate whether $C_{44}$Mab-108 can be used for immunohistochemical analyses using paraffin-embedded tumor sections, we used sequential sections of an OSCC tissue microarray. In a well-differentiated OSCC section (Figure 4A–H), a clear membrane-staining in OSCC was observed by $C_{44}$Mab-46 (Figure 4C,D), but hardly detected by $C_{44}$Mab-108 (Figure 4A,B). In an OSCC section with the stromal invaded phenotype (Figure 4I–P), $C_{44}$Mab-108 (Figure 4I,J) strongly stained stromal invaded OSCC and could clearly distinguish tumor cells from stromal tissues. In contrast, $C_{44}$Mab-46 (Figure 4K,L) stained both. The reactivity of $C_{44}$Mab-108 was eliminated completely by a CD44 peptide (aa 271–290), which contains the $C_{44}$Mab-108 epitope (Figure 4Q,R). We summarized the data of immunohistochemical analysis in Table 1; $C_{44}$Mab-108 stained 20 out of 50 (40%)

cases of OSCC. Similar staining patterns were also observed in esophageal SCC tissues. $C_{44}$Mab-108 also stained tumors selectively, and could clearly distinguish tumor cells from stroma (Supplementary Figure S2). These results indicated that $C_{44}$Mab-108 is useful for immunohistochemical analysis of paraffin-embedded tumor sections.

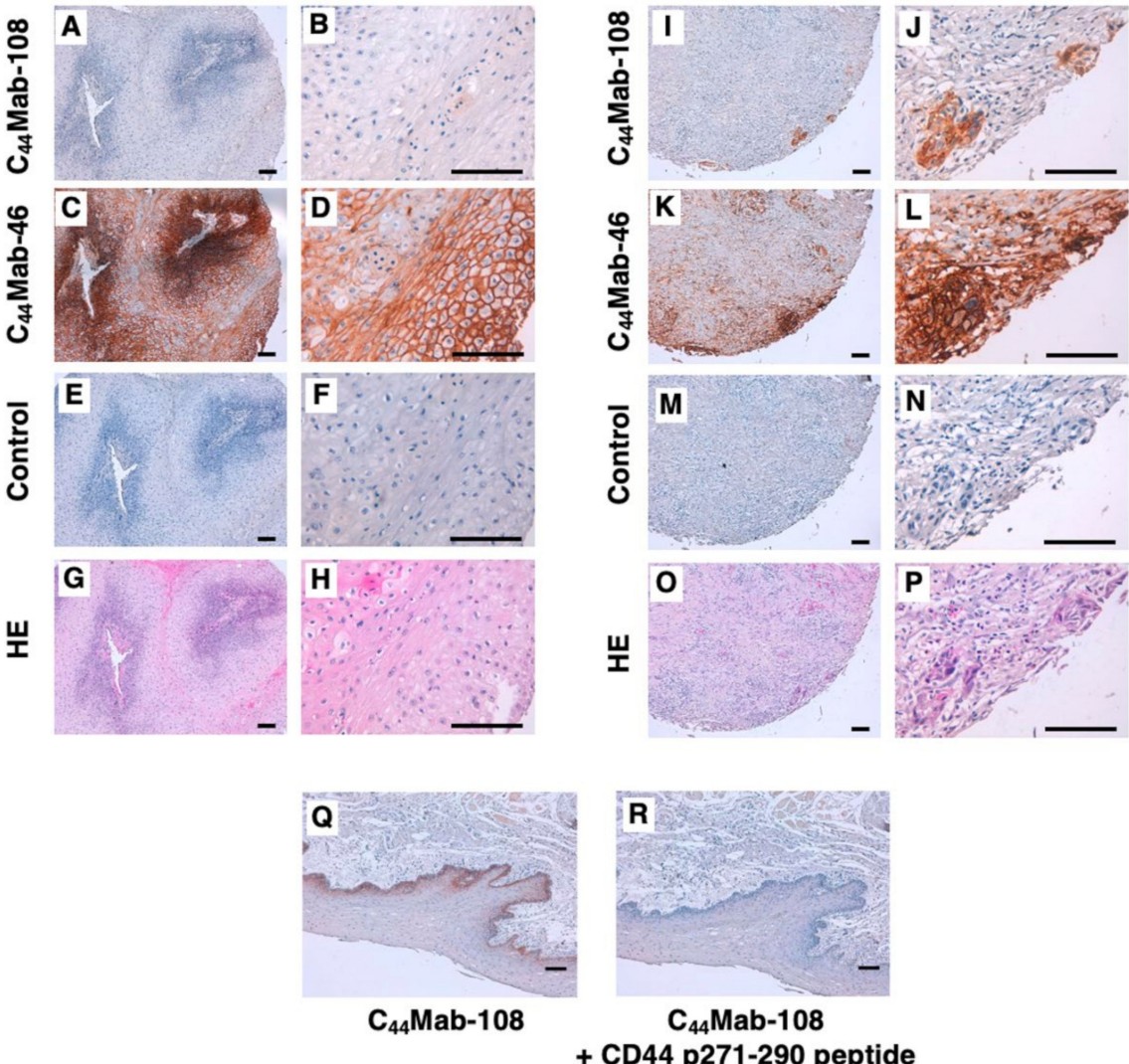

**Figure 4.** Immunohistochemical analysis using $C_{44}$Mab-108 and $C_{44}$Mab-46 against oral squamous cell carcinoma (OSCC) tissues. After antigen retrieval, the sections were incubated with 10 µg/mL of $C_{44}$Mab-108 (**A,B,I,J**), 1 µg/mL of $C_{44}$Mab-46 (**C,D,K,L**), and without the primary antibody (control) (**E,F,M,N**) followed by treatment with the Envision+ kit. The color was developed using 3,3′-diaminobenzidine tetrahydrochloride (DAB), and the sections were counterstained with hematoxylin. (**G,H,O,P**) Hematoxylin and eosin (HE) staining. (**Q,R**) Blocking of the $C_{44}$Mab-108 reactivity to OSCC tissue by the CD44 peptide (aa 271–290) containing the $C_{44}$Mab-108 epitope. After antigen retrieval, sections were incubated with $C_{44}$Mab-108 (10 µg/mL) or $C_{44}$Mab-108 (10 µg/mL) plus human CD44 peptide (aa 271–290, 10 µg/mL) followed by treatment with the Envision+ kit. The color was developed using DAB, and sections were counterstained with hematoxylin. Scale bar = 100 µm.

**Table 1.** Immunohistochemical analysis using $C_{44}$Mab-108 against OSCC tissues.

| No. | Age | Sex | Organ/Anatomic Site | Pathology Diagnosis | TNM | $C_{44}$Mab-108 | $C_{44}$Mab-46 |
|---|---|---|---|---|---|---|---|
| 1 | 78 | M | Tongue | Squamous cell carcinoma of tongue | T2N0M0 | - | + |
| 2 | 40 | M | Tongue | Squamous cell carcinoma of tongue | T2N0M0 | - | ++ |
| 3 | 75 | F | Tongue | Squamous cell carcinoma of tongue | T2N0M0 | ++ | + |
| 4 | 35 | F | Tongue | Squamous cell carcinoma of tongue | T2N0M0 | + | +++ |
| 5 | 61 | M | Tongue | Squamous cell carcinoma of tongue | T2N0M0 | + | +++ |
| 6 | 41 | F | Tongue | Squamous cell carcinoma of tongue | T2N0M0 | + | ++ |
| 7 | 64 | M | Tongue | Squamous cell carcinoma of right tongue | T2N2M0 | - | + |
| 8 | 76 | M | Tongue | Squamous cell carcinoma of tongue | T1N0M0 | - | ++ |
| 9 | 50 | F | Tongue | Squamous cell carcinoma of tongue | T2N0M0 | + | ++ |
| 10 | 44 | M | Tongue | Squamous cell carcinoma of tongue | T2N1M0 | - | +++ |
| 11 | 53 | F | Tongue | Squamous cell carcinoma of tongue | T1N0M0 | - | +++ |
| 12 | 46 | F | Tongue | Squamous cell carcinoma of tongue | T2N0M0 | - | ++ |
| 13 | 50 | M | Tongue | Squamous cell carcinoma of root of tongue | T3N1M0 | + | ++ |
| 14 | 36 | F | Tongue | Squamous cell carcinoma of tongue | T1N0M0 | + | +++ |
| 15 | 63 | F | Tongue | Squamous cell carcinoma of tongue | T1N0M0 | - | ++ |
| 16 | 46 | M | Tongue | Squamous cell carcinoma of tongue | T2N0M0 | + | ++ |
| 17 | 58 | M | Tongue | Squamous cell carcinoma of tongue | T2N0M0 | + | ++ |
| 18 | 64 | M | Lip | Squamous cell carcinoma of lower lip | T1N0M0 | - | +++ |
| 19 | 57 | M | Lip | Squamous cell carcinoma of lower lip | T2N0M0 | - | +++ |
| 20 | 61 | M | Lip | Squamous cell carcinoma of lower lip | T1N0M0 | - | +++ |
| 21 | 60 | M | Gum | Squamous cell carcinoma of gum | T3N0M0 | + | ++ |
| 22 | 60 | M | Gum | Squamous cell carcinoma of gum | T1N0M0 | + | +++ |
| 23 | 69 | M | Gum | Squamous cell carcinoma of upper gum | T3N0M0 | - | +++ |
| 24 | 53 | M | Bucca cavioris | Squamous cell carcinoma of bucca cavioris | T2N0M0 | - | + |
| 25 | 55 | M | Bucca cavioris | Squamous cell carcinoma of bucca cavioris | T1N0M0 | - | ++ |
| 26 | 58 | M | Tongue | Squamous cell carcinoma of base of tongue | T1N0M0 | - | ++ |
| 27 | 63 | M | Oral cavity | Squamous cell carcinoma | T1N0M0 | - | ++ |
| 28 | 48 | F | Tongue | Squamous cell carcinoma of tongue | T1N0M0 | - | ++ |
| 29 | 80 | M | Lip | Squamous cell carcinoma of lower lip | T1N0M0 | - | +++ |
| 30 | 77 | M | Tongue | Squamous cell carcinoma of base of tongue | T2N0M0 | - | +++ |
| 31 | 59 | M | Tongue | Squamous cell carcinoma of tongue | T2N0M0 | - | ++ |
| 32 | 77 | F | Tongue | Squamous cell carcinoma of tongue | T1N0M0 | + | +++ |
| 33 | 56 | M | Tongue | Squamous cell carcinoma of root of tongue | T2N1M0 | - | ++ |
| 34 | 60 | M | Tongue | Squamous cell carcinoma of tongue | T2N1M0 | + | +++ |
| 35 | 62 | M | Tongue | Squamous cell carcinoma of tongue | T2N0M0 | ++ | +++ |
| 36 | 67 | F | Tongue | Squamous cell carcinoma of tongue | T2N0M0 | ++ | +++ |
| 37 | 47 | F | Tongue | Squamous cell carcinoma of tongue | T2N0M0 | + | +++ |
| 38 | 37 | M | Tongue | Squamous cell carcinoma of tongue | T2N1M0 | - | - |
| 39 | 55 | F | Tongue | Squamous cell carcinoma of tongue | T2N0M0 | - | ++ |
| 40 | 56 | F | Bucca cavioris | Squamous cell carcinoma of bucca cavioris | T2N0M0 | + | +++ |
| 41 | 49 | M | Bucca cavioris | Squamous cell carcinoma of bucca cavioris | T1N0M0 | - | - |
| 42 | 45 | M | Bucca cavioris | Squamous cell carcinoma of bucca cavioris | T2N0M0 | - | ++ |
| 43 | 42 | M | Bucca cavioris | Squamous cell carcinoma of bucca cavioris | T3N0M0 | - | +++ |
| 44 | 44 | M | Jaw | Squamous cell carcinoma of right drop jaw | T1N0M0 | + | +++ |
| 45 | 40 | F | Tongue | Squamous cell carcinoma of base of tongue | T2N0M0 | - | +++ |
| 46 | 49 | M | Bucca cavioris | Squamous cell carcinoma of bucca cavioris | T1N0M0 | + | +++ |
| 47 | 56 | F | Tongue | Squamous cell carcinoma of base of tongue | T2N0M0 | - | + |
| 48 | 42 | M | Bucca cavioris | Squamous cell carcinoma of bucca cavioris | T3N0M0 | + | +++ |
| 49 | 87 | F | Face | Squamous cell carcinoma of left face | T2N0M0 | - | + |
| 50 | 50 | M | Gum | Squamous cell carcinoma of gum | T2N0M0 | - | ++ |

-, No stain; +, Weak intensity; ++, Moderate intensity; +++, Strong intensity.

## 4. Discussion

CD44v have been implicated as a marker of cancer-initiating cells and plays pro-tumorigenic functions in many carcinomas [37]. The v3-encoded region possesses heparan sulfate moieties that recruit to HB-EGF and FGFs [12]. The v6-encoded region forms a ternary complex with MET and HGF, which is essential for the c-MET activation [13]. Furthermore, the CD44 intracellular domain is required for the recruitment of the ezrin, radixin, and moesin complex, which potentiates the c-MET signal transduction [38]. These functions of CD44v are important for the malignant progression of tumors to potentiate proliferation, invasiveness, and metastatic spread. Furthermore, CD44v interacts with a cystine–glutamate transporter (xCT) subunit through a v8-10-encoded region [14], which mediates oxidative stress and antitumor drug resistance in several carcinomas [39]. Therefore, the establishment and characterization of mAbs, which recognize CD44v are thought to be essential for the development of CD44-targeting tumor diagnosis and therapy.

In contrast, the function of the v4-encoded region has not been elucidated. Therefore, specific antibodies against CD44v4 have been desired. In this study, we developed $C_{44}$Mab-108, which can recognize the v4 region. We showed the usefulness of flow cytometry (Figure 2), western blotting (Figure 3), and immunohistochemistry (Figure 4 and Supplementary Figure S2). In Supplementary Figure S3, we showed the homology of the v4 region between human, mouse, rat, and Chinese hamster sequences. In our preliminary epitope mapping analysis, the alanine-substitution of each amino acid in the human v4 region, which is identical to the mouse sequence, reduced the reactivity of $C_{44}$Mab-108 (manuscript preparation). The result suggests that $C_{44}$Mab-108 recognizes both human and mouse CD44v4.

As shown in Figure 2C, the affinity of $C_{44}$Mab-108 for CHO/CD44v3-10 was not high. Although high affinity is the general goal for antibody generation, recent reports showed that low rather than high affinity exhibits elevated activity through inducing clustering of receptors. For instance, low-affinity variants of an anti-PD-1 mAb nivolumab mediated more potent signaling and affected T-cell activation. These findings reveal a new paradigm for antibody-mediated receptor signaling [40]. Since CD44 is involved in intracellular signaling, the relationship between the antibody affinity and the effect of cellular signaling should be investigated in future studies.

Most CD44v mAbs have been developed by the immunization of recombinant or cell surface expressed CD44v, which received the glycosylation. CD44 is known to be heavily glycosylated [41], and the glycosylation pattern is thought to depend on the host cells. Moreover, it is not clear whether the available anti-CD44 mAbs recognize the peptide or glycopeptide structures of CD44v. Since $C_{44}$Mab-108 was established by the peptide immunization, $C_{44}$Mab-108 can recognize the definite peptide structure of the variant 4-encoded region. Since the region is also expected to receive the glycosylation [41], further studies are required to reveal whether the glycosylation affects the recognition by $C_{44}$Mab-108. In the immunohistochemical analysis (Figure 4, Table 1), $C_{44}$Mab-108 could detect endogenous CD44v4 in some OSCC tissues. Although an anti-pan-CD44 mAb, $C_{44}$Mab-46 recognized not only OSCC tissues, but also stromal tissues, $C_{44}$Mab-108 stained the tumor tissues selectively. Furthermore, $C_{44}$Mab-108 recognized a limited population of OSCC tissues compared to $C_{44}$Mab-46 (Figure 4, Table 1). In the future, we should evaluate the clinical significance of the expression pattern and histology of $C_{44}$Mab-108-positive OSCC. The information could help the future diagnostic and therapeutic applications of $C_{44}$Mab-108 in OSCC.

Clinical trials of anti-CD44 mAbs have been conducted [42]. A humanized anti-CD44v6 mAb BIWA4 (bivatuzumab)−mertansine drug conjugate was evaluated but discontinued due to severe skin toxicities [43,44]. An anti-pan CD44 mAb, RG7356, exhibited an acceptable safety profile in patients with advanced CD44-expressing solid tumors. However, the study was terminated due to no evidence of a clinical and pharmacodynamic dose-response relationship with RG7356 [45]. Therefore, the development of anti-CD44 mAbs with more potent and fewer side effects is required.

We previously established cancer-specific mAbs (CasMabs) against podoplanin [46–48] and podocalyxin [49], which are expressed in many cancers including OSCC [50]. The strategy also contributes to the development of various types of mAbs [51–58]. An anti-podoplanin CasMab, LpMab-2 can recognize the cancer-type aberrant glycosylation of Thr55 and/or Ser56 with surrounding peptide, which is not expressed in normal cells [48]. LpMab-2 has also been applied to chimeric antigen receptor T-cell therapy in mice models [59]. We should understand the cancer-type glycosylation of CD44v, which could support the development of cancer-specific anti-CD44v mAbs.

**Supplementary Materials:** The following supporting information can be downloaded at: https://www.mdpi.com/article/10.3390/cimb45030121/s1, Table S1, Determination of C$_{44}$Mab-108 isotype by ELISA. Figure S1, Conformation of C$_{44}$Mab-108 epitope by ELISA using peptides which cover the CD44v3-10 extracellular domain. Figure S2, Immunohistochemical analysis using C$_{44}$Mab-108 and C$_{44}$Mab-46 against esophageal squamous cell carcinoma tissues. Figure S3, An alignment of human, mouse, rat, and Chinese hamster (Cham) CD44 variant 4 sequences.

**Author Contributions:** H.S., N.G. and T.T. performed the experiments. M.K.K. and Y.K. designed the experiments. H.S., N.G. and Y.K. analyzed the data. H.S. and Y.K. wrote the manuscript. All authors have read and agreed to the published version of the manuscript.

**Funding:** This research was supported in part by Japan Agency for Medical Research and Development (AMED) under Grant Numbers: JP22ama121008 (to Y.K.), JP22am0401013 (to Y.K.), JP22bm1004001 (to Y.K.), JP22ck0106730 (to Y.K.), and JP21am0101078 (to Y.K.), and by the Japan Society for the Promotion of Science (JSPS) Grants-in-Aid for Scientific Research (KAKENHI) grant nos. 21K20789 (to T.T.), 22K06995 (to H.S.), 21K07168 (to M.K.K.), and 22K07224 (to Y.K.).

**Institutional Review Board Statement:** The animal study protocol was approved by the Animal Care and Use Committee of Tohoku University (Permit number: 2019NiA-001) for studies involving animals.

**Data Availability Statement:** The data presented in this study are available in the article and supplementary material.

**Acknowledgments:** The authors would like to thank Saori Okuno and Saori Handa (Department of Antibody Drug Development, Tohoku University Graduate School of Medicine) for technical assistance.

**Conflicts of Interest:** The authors have no conflict of interest to declare.

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
