# Peer review of "Development of a Novel Anti-CD44 Variant 4 Monoclonal Antibody C44Mab-108 for Immunohistochemistry"

_cimb, doi:10.3390/cimb45030121_

Round 1

Reviewer 1 Report

Hiroyuki Suzuki and colleagues developed a monoclonal antibody against human CD44 variant 4 and also tested its usefulness in flow cytometry, western blotting, and immunohistochemistry. Overall, the study is well-designed, and the experimental approaches and the interpretation of the data are appropriate.

I only have a few questions here. 

1.    From figure 2A and C, C44Mab-108 showed moderate affinity, not high. What do you think made it? Will it affect the further use the author mentioned in the Discussion?

2.    How did you know the antibody is IgG1, kappa? Please add the figure and the methods.

3.    What’s the role of KYSE70 and KYSE770?

4.  The authors use P3U1 cell line for fusion. What’s the advantage compared to SP2/0?

Author Response

Hiroyuki Suzuki and colleagues developed a monoclonal antibody against human CD44 variant 4 and also tested its usefulness in flow cytometry, western blotting, and immunohistochemistry. Overall, the study is well-designed, and the experimental approaches and the interpretation of the data are appropriate.

I only have a few questions here.

  1. From figure 2A and C, C44Mab-108 showed moderate affinity, not high. What do you think made it? Will it affect the further use the author mentioned in the Discussion?

Although high affinity is routinely the goal for therapeutic antibody generation, recent report showed that low rather than high affinity delivers greater activity through increased clustering. This approach delivered higher immune cell activation, in vivo T cell expansion and antitumor activity in the case of CD40. Furthermore, low-affinity variants of the clinically important antagonistic anti-PD-1 monoclonal antibody nivolumab also mediated more potent signaling and affected T cell activation. These findings reveal a new paradigm for antibody-mediated receptor signaling (Nature, 614, 539-, 2023). Since CD44 involves in the intracellular signaling, the relationship between the antibody affinity and the effect of cellular signaling should be investigated.

We added above in discussion.

  1. How did you know the antibody is IgG1, kappa? Please add the figure and the methods.

We added the method in the text and data as supplemental Table S1.

3.What’s the role of KYSE70 and KYSE770?

Since we could detect CD44v4 expression in the immunohistochemistry of esophageal SCC tissue (supplementary Fig. S2), we chose esophageal SCC cell lines, KYSE70 and KYSE770 for western blot analysis.

4.The authors use P3U1 cell line for fusion. What’s the advantage compared to SP2/0?

We have used P3U1 in many antibody productions, and have established protocols. We would like to do a comparison with SP2/0 in the future.

Reviewer 2 Report

In the current manuscript 'Development of a novel anti-CD44 variant 4 monoclonal antibody C44Mab-108 for immunohistochemistry', H. Suzuki et al. isolated one CD44 variant 4 specific antibody, C44Mab-108, and the following assays including WB, flow, IHC all showed this Ab can recognize the CD44 variant specifically with a high affinity. This manuscript shows the potential application of C44Mab-108 in the diagnosis.

The current manuscript meets the publication criteria, and recommend to publish in the present format.

Author Response

In the current manuscript 'Development of a novel anti-CD44 variant 4 monoclonal antibody C44Mab-108 for immunohistochemistry', H. Suzuki et al. isolated one CD44 variant 4 specific antibody, C44Mab-108, and the following assays including WB, flow, IHC all showed this Ab can recognize the CD44 variant specifically with a high affinity. This manuscript shows the potential application of C44Mab-108 in the diagnosis.

The current manuscript meets the publication criteria, and recommend to publish in the present format.

Thank you very much for positive comment.